

# Mitochondrial sequence data reveal population structure within *Pustulosa pustulosa*

David Rodriguez[1], Stephen F. Harding[1], Shashwat Sirsi[1],
Kelly McNichols-O'Rourke[2], Todd Morris[2], Michael R. J. Forstner[1] and
Astrid N. Schwalb[1]

[1] Department of Biology, Texas State University, San Marcos, TX, United States
[2] Great Lakes Laboratory for Fisheries and Aquatic Sciences, Fisheries and Oceans Canada, Burlington, ON, Canada

## ABSTRACT

Unionid mussels are among the most imperiled group of organisms in North America, and *Pustulosa pustulosa* is a freshwater species with a relatively wide latitudinal distribution that extends from southern Ontario, Canada, to Texas, USA. Considerable morphological and geographic variation in the genus *Pustulosa* (formerly *Cyclonaias*) has led to uncertainty over species boundaries, and recent studies have suggested revisions to species-level classifications by synonymizing *C. aurea*, *C. houstonensis*, *C. mortoni*, and *C. refulgens* with *C. pustulosa* (currently *P. pustulosa*). Owing to its wide range and shallow phylogenetic differentiation, we analyzed individuals of *P. pustulosa* using mitochondrial DNA sequence data under a population genetics framework. We included 496 individuals, which were comprised of 166 samples collected during this study and 330 additional sequences retrieved from GenBank. Pairwise $\Phi_{ST}$ measures based on ND1 data suggested there may be up to five major geographic groups present within *P. pustulosa*. Genetic differentiation between regions within Texas was higher compared to populations from the Mississippi and Great Lakes populations, which may reflect differences in historical connectivity. Mitochondrial sequence data also revealed varying demographic histories for each major group suggesting each geographic region has also experienced differential population dynamics in the past. Future surveys should consider exploring variation within species after phylogeographic delimitation has been performed. In this study, we begin to address this need for freshwater mussels *via* the *P. pustulosa* system.

## INTRODUCTION

Unionid mussels (order: Unionida) belong to one of the most imperiled group of organisms in North America with two-thirds of the 298 currently recognized species being either threatened, vulnerable, endangered, or presumed/possibly extinct (*Lopes-Lima et al., 2018, 2014; Williams et al., 2017*). Approximately 50 species occur in Texas, USA, of which 15 are considered threatened (Texas Administrative Code, 31 TAC §65.175), with one

Corresponding author
David Rodriguez, drdz@txstate.edu

species federally listed as endangered (Federal Register, 2015-32284). A crucial step in conservation and management of any species is the examination of genetic variation and partitioning in relation to its geographic distribution. Recently, a revised list of freshwater mussels included changes in their taxonomy at the species level (*Lopes-Lima et al., 2019*; *Williams et al., 2017*). However, once species are delimited, measuring intraspecific variation and population structure is then needed to better tailor management efforts to regional conditions.

Among recent taxonomic changes were the re-assignment of nine *Quadrula* (Rafinesque, 1820) species to the genus *Cyclonaias* (*Lopes-Lima et al., 2019*; *Williams et al., 2017*), and more recently, several of those taxa were subsequently assigned to *Pustulosa* (*Neemuchwala et al., 2023*). The focal species of this study, *Pustulosa* (*Cyclonaias*) *pustulosa*, has a wide latitudinal distribution ranging from southern Ontario, Canada to Texas, USA. Considerable phenotypic plasticity and intrinsic shell variation could be partly responsible for taxonomic confusion within and among species of *Quadrula*, *Cyclonaias*, and *Pustulosa* (*Campbell et al., 2005*; *Neemuchwala et al., 2023*; *Serb, Buhay & Lydeard, 2003*). Some mussels occurring in different states of the USA (Texas, Louisiana, and Mississippi) are difficult to distinguish morphologically, and a recent revision at the species-level suggested synonymizing four species (*C. aurea*, *C. houstonensis*, *C. mortoni*, and *C. refulgens*) as *C. pustulosa* (currently *P. pustulosa*) based on analysis of genetic and morphological data (*Johnson et al., 2018*; *Lopes-Lima et al., 2019*); albeit, some weak associations between phylogenetic diversity and geography have also been detected by the same studies.

These previously detected genetic and geographic associations may represent undescribed intraspecific population structure within *P. pustulosa* sensu *lato*, which we consider to include the four previously recognized nominal species and *P. pustulosa*. An investigation of the population genetics of *P. pustulosa* sensu *stricto* (populations previously recognized as *Cyclonaias pustulosa*) has been performed within the St. Croix River (*Szumowski et al., 2012*); however, this was before the recent taxonomic changes mentioned above, and the study was restricted to a few local populations. Thus, owing to weak phylogenetic resolution detected in previous studies (*Johnson et al., 2018*; *Lopes-Lima et al., 2019*), a treatment of *P. pustulosa* sensu *lato* within a population genetics framework is still needed.

The concepts of adaptive evolutionary conservation and delimitation of evolutionarily significant units (ESUs; *Ryder, 1986*) have been promoted as objective strategies to preserve genetic diversity and evolutionary potential (*Fraser & Bernatchez, 2001*). This process encompasses more than identifying reciprocally monophyletic groups and takes into account both the ecology and evolutionary biology of a target taxon (*Fraser & Bernatchez, 2001*; *Grant, 1995*; *Moritz, 1994*). Related to ESUs are regional management units (RMUs), which have been used to organize populations of wide-ranging species, above local populations but below the species level, with potentially independent evolutionary trajectories (*Wallace et al., 2010*). Thus, the collapsing of separate species into one wide-ranging species owing to minimal phylogenetic resolution, as in *P. pustulosa* sensu *lato*, can affect subsequent management decisions for independently evolving populations

(*Berg & Berg, 2000*; *Mulvey et al., 1997*). Therefore, conservation of potentially threatened taxa requires the consolidation of all available data to designate management units within ESUs after species delimitation has been performed.

Our goal was to examine genetic variation, patterns of isolation by distance, and the demographic history of *P. pustulosa* through a large part of its range using mitochondrial DNA but with a focus on Texas populations, where *P. pustulosa* was previously identified as *C. aurea*, *C. houstonensis*, and *C. mortoni* in different parts of the state. While there is phylogenetic support for synonymy of these species (*Johnson et al., 2018*; *Lopes-Lima et al., 2019*), evidence of population structure and different demographic histories for each population across this wide latitudinal range would warrant RMUs to preserve the independent evolutionary trajectory of each distinct regional population. Alternatively, no evidence of regional population structure would suggest panmixia and support one all-encompassing management unit.

## MATERIALS AND METHODS

### Sample collection

We sampled mussels from five river basins in Texas, USA (Colorado, Guadalupe, Neches, San Antonio, and Trinity) and three basins in Ontario, Canada (Sydenham, Grand, and Thames) from 2015 to 2018 (Fig. 1). We collected DNA primarily by swabbing the foot using a sterile cytology brush with a polystyrene handle. After swabbing, the brush was immediately placed into 1.5 mL screw cap tubes filled with 95% EtOH and stored at −20 °C until extraction. We transported a subset of mussels to the San Marcos Aquatic Resources Center or Texas State University, San Marcos, Texas, USA and maintained them in experimental mesocosms. When captive mussels expired, we immediately preserved them in 95% EtOH and stored the carcasses at −20 °C to serve as reference tissue samples. From those preserved tissue samples, we took 10–20 mg of tissue from the foot or mantle. At the start of the study, we extracted DNA from swab or tissue using the Gentra PureGene DNA extraction kit (Qiagen, Hilden, Germany) following the manufacturer's recommendations. However, in an attempt to increase yield, we used the GeneJET Genomic DNA Purification Kit (Thermo Scientific, Waltham, MA, USA) for the remainder of the project.
We diagnosed all genomic DNA purifications for the presence of high molecular weight DNA using a 1% agarose gel and 0.5X TBE buffer solution. Field collections were approved by Texas Parks and Wildlife Department (Scientific Permit Number SPR-1014-236) and Canada Department of Fisheries and Oceans (Permit 15-PCAA-00009).

### DNA sequencing

To compensate for variable DNA yields from non-lethal swabs and to increase sample size, we performed a nested PCR protocol using primers targeting the mitochondrial NADH dehydrogenase 1 gene (ND1). We focused on this marker to leverage the available reference data for other populations in GenBank (Appendix S1). The initial PCR included primers Leu-uurF (5′-TGG CAG AAA AGT GCA TCA GAT TAA AGC-3′) and NIJ-12073 (5′-TCG GAA TTC TCC TTC TGC AAA GTC-3′) (*Serb, Buhay & Lydeard, 2003*), while the second PCR used Leu-uurF and Quadrula_NDlintR (5′-TGG AGC TCG GTT

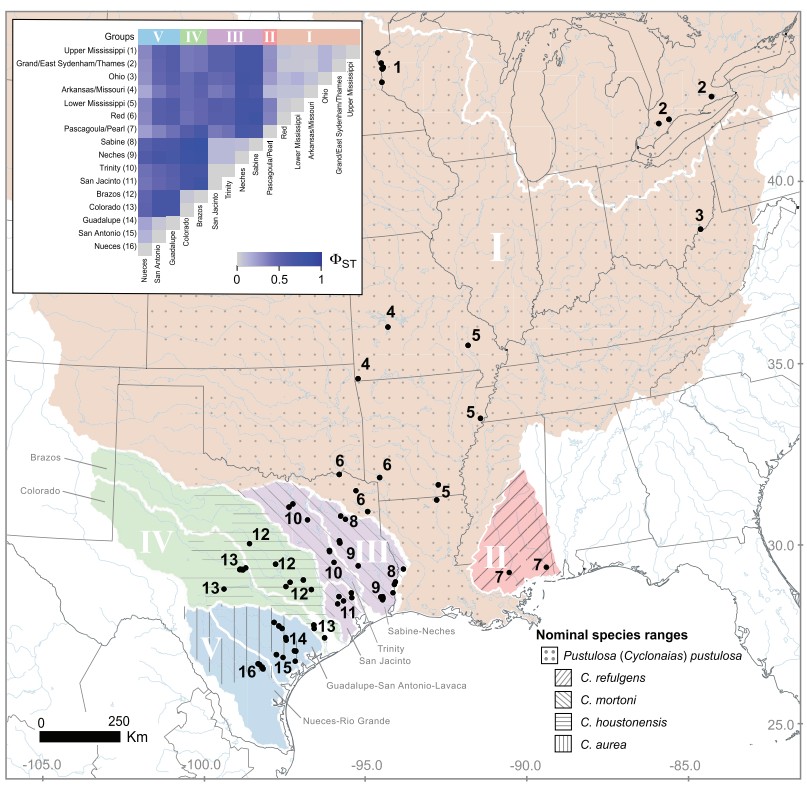

**Figure 1 Geographic distribution of *Pustulosa* (*Cyclonaias*) *pustulosa* samples.** The inset shows pairwise measures of $\Phi_{ST}$ based on ND1 haplotype frequencies among *P. pustulosa* populations from different river drainages (numbered). Major groups, based on $\Phi_{ST}$ and geography (river basin) are designated with Roman numerals. Within Texas, large river basins are outlined (white). Approximate ranges for historically recognized nominal species within *P. pustulosa* are also shown with hatching (*Lopes-Lima et al., 2019*).

TGT TTC TGC CA-3′), which was designed for this study. Initial PCRs comprised of 25 µL total volumes containing 12.5 µL DreamTaq PCR Master Mix 2X (ThermoFisher, Inc., Waltham, MA, USA), 1 µM forward primer, 1 µM reverse primer, 0.5 µM MgCl$_2$, ~1–10 ng of genomic DNA, and nuclease-free H$_2$O to volume. The second PCR contained half volumes of all reagents and 0.25 to 0.5 µL of PCR product, depending on the relative concentration of amplicons assessed *via* band intensity after gel electrophoresis. The thermal profile for both reactions followed an initial denaturation at 95 °C for 5 min; then 34 cycles of 95 °C for 1 min., 55 °C for 1 min., and 72 °C for 1 min. 50 s; and a final elongation at 72 °C for 7 min.

To enzymatically digest unincorporated dNTPs and primers in the completed PCR reactions, we used ExoSAP-IT® (Affymetrix, Santa Clara, CA, USA). We cycle sequenced both DNA strands independently using BigDye v3.1 (Applied Biosystems Inc., Waltham, MA, USA), which we purified using Sephadex G-50 Superfine (GE Healthcare) gel filtration medium. DNA sequencing was performed on an Applied Biosystems 3500 Genetic Analyzer. We used Geneious 9.1.8 (Biomatters Ltd., Auckland, New Zealand) to edit and assemble bidirectional chromatograms and to build and trim alignments, and we accessioned all newly generated sequence data into GenBank (OQ656460–OQ656625).

**Table 1 Partitioning schemes for analysis of molecular variance tests.**

| Basin | Scenario A | Scenario B | N | h |
|---|---|---|---|---|
| Upper Mississippi | *C. pustulosa* | Mississippi embayment | 66 | 42 |
| Great Lakes[a] | *C. pustulosa* | Mississippi embayment | 43 | 22 |
| Ohio | *C. pustulosa* | Mississippi embayment | 9 | 9 |
| Arkansas/Missouri | *C. pustulosa* | Mississippi embayment | 8 | 8 |
| Lower Mississippi | *C. pustulosa* | Mississippi embayment | 31 | 27 |
| Red | *C. pustulosa* | Mississippi embayment | 28 | 21 |
| Pascagoula/Pearl[b] | Not included | Not included | 10 | 8 |
| Sabine | *C. mortoni* | Sabine-Neches | 8 | 7 |
| Neches | *C. mortoni* | Sabine-Neches | 61 | 26 |
| Trinity | *C. mortoni* | Trinity-San Jacinto | 18 | 14 |
| San jacinto | *C. mortoni* | Trinity-San Jacinto | 9 | 8 |
| Brazos | *C. houstonensis* | Central Texas | 18 | 7 |
| Colorado | *C. houstonensis* | Central Texas | 58 | 17 |
| Guadalupe | *C. aurea* | Central Texas | 66 | 42 |
| San antonio | *C. aurea* | Central Texas | 22 | 13 |
| Nueces | *C. aurea* | Rio Grande | 41 | 12 |

Notes:
[a] Grand, East Sydenham, and Thames Rivers.
[b] Populations previously described as *C. refulgens*.
Analyses are based on ND1 sequence data from *Pustulosa (Cyclonaias) pustulosa* based on historical nominal species (Scenario A) or unionid biogeographical provinces (Scenario B), N, sample size; and h, number of haplotypes.

Previously published sequences showing uncorrected sequence similarity greater than 96% regardless of named taxon were also incorporated into the downstream analyses; however, we excluded any sequences with ambiguities (Appendix S1).

## Estimating population differentiation

To determine range-wide population structure based on ND1 sequences, we used Arlequin version 3.5.2.2 (*Excoffier & Lischer, 2010*) and calculated measures of haplotype diversity and estimated pairwise values of $\Phi_{ST}$ between 16 river drainages by computing a distance matrix of raw pairwise differences. We inferred the significance of pairwise comparisons with an exact test (1,000 iterations) and adjusted the resulting *P*-values *via* Bonferroni correction. To evaluate haplotype sharing among populations, we visualized haplotype diversity and abundance in a median-joining network using POPART (*Bandelt, Forster & Rohl, 1999*; *Leigh & Bryant, 2015*) and by arbitrarily enumerating haplotypes and plotting haplotype frequencies. We excluded populations from the Pascagoula/Pearl rivers from downstream analyses owing to low sample sizes.

We compared two *a priori*, biologically relevant hierarchical population subdivision scenarios using AMOVA tests as implemented in Arlequin with 1,023 permutations (Table 1). Populations (*i.e.*, rivers) were either grouped according to the ranges of previously described species now comprising *P. pustulosa* sensu *lato* or grouped by unionid biotic provinces. Specifically, six provinces (Great Plains, Mississippi Embayment, Sabine and Neches drainages, Trinity and San Jacinto drainages, Central Texas, and the Rio

Grande drainage) were previously delineated using a combination of ecological and genetic data (see *de Moulpied et al., 2022*). We sought to infer which scenario maximized the percentage of genetic variation among groups (*i.e.*, FCT).

### Inferring demographic histories

To infer the demographic history of each group using mitochondrial haplotype data, we estimated values for Tajima's D and performed mismatch distribution analyses under a model of spatial expansion in Arlequin. We constructed a plot using the observed number of site differences between sequence pairs and the expected distribution under a constant population size model.

### Isolation by distance (IBD)

We performed analyses and plotting in R version 4.2.2 (*R Core Team, 2023*) to determine if population structure was explained by geographic distance at the individual level. We constructed a Haversine distance matrix among all individuals using the *geodist* (geodist) function (*Padgham, 2021*) and calculated a raw genetic distance matrix using *dist. dna* (ape) (*Paradis & Schliep, 2019*). To test for IBD *via* a Mantel test, we used the two resulting matrices as input for the *mantel* (vegan) function (*Oksanen et al., 2022*) and set the number of permutations to 1,000. We used *kde2d* (MASS) (*Venables & Ripley, 2002*) to overlay a two-dimensional density plot over a scatterplot of pairwise genetic and geographic distances.

## RESULTS

We generated ND1 sequence data for individuals collected in Texas, USA and Ontario, Canada and incorporated publicly available data from other studies (Appendix S1). Altogether, geographic sampling spanned approximately 17.50 degrees of latitude and encompassed a large part of the range of *P. pustulosa* sensu *lato* (Fig. 1). A 528 base pair alignment was assembled for 496 individuals at the ND1 locus with no missing data or base ambiguities. Overall, the *P. pustulosa* group showed high haplotype diversity (H = 229) across 163 polymorphic sites (S).

### Signatures of population differentiation

Estimates of pairwise $\Phi_{ST}$ among river drainages showed partitioning of ND1 variation according to geography (Fig. 1 inset, Table S1), all together, values ranged from 0 to 0.85. Collectively, Mississippi basin populations, including the Red River showed less subdivision among river drainages ($\Phi_{ST}$ 0 to 0.11) than populations within Texas ($\Phi_{ST}$ 0 to 0.85). The among group variation was higher (47.7%) when rivers were grouped by historical nominal species groups (Scenario A), and lower when rivers were grouped by unionid biotic provinces (Scenario B; 26.9%) (Table 2).

Based on the population-level genetic partitioning results, individuals from drainages within the Mississippi basin and the Great Lakes were placed into Group I; individuals from the Pascagoula/Pearl populations were placed into Group II; individuals from the Sabine, Neches, Trinity, and San Jacinto rivers were placed into Group III; individuals from the Brazos and Colorado were placed into Group IV; individuals from the

**Table 2 Analysis of molecular variance results for different hierarchical partitioning scenarios of _Pustulosa_ (_Cyclonaias_) _pustulosa_ populations.**

| Source of variation | df | % variation | F statistics | P val |
|---|---|---|---|---|
| **Scenario A** | | | | |
| Among groups | 3 | 47.7 | FCT = 0.48 | 0.000 |
| Among populations within groups | 11 | 3.1 | FSC = 0.07 | 0.000 |
| Within populations | 471 | 49.2 | FST = 0.50 | 0.000 |
| **Scenario B** | | | | |
| Among groups | 4 | 26.9 | FCT = 0.27 | 0.003 |
| Among populations within groups | 10 | 21.8 | FSC = 0.30 | 0.000 |
| Within populations | 471 | 51.3 | FST = 0.48 | 0.000 |

Note:
Namely, (A) by historical nominal species ranges (_i.e._, _Cyclonaias pustulosa_, _C. refulgens_, _C. mortoni_, C. houstonensis, and C. aurea) (_Johnson et al., 2018_, _Lopes-Lima et al., 2019_), and (B) by unionid biogeographical provinces (_de Moulpied et al., 2022_) (see Table 1). Analyses are based on 528 base pairs of the ND1 mitochondrial gene.

Guadalupe, San Antonio, and Nueces rivers were placed into Group V; (Fig. 1). Plotted haplotype frequencies and haplotype networks showed minimal haplotype sharing between these defined groups (Fig. 2) with haplotypes generally clustering together in the haplotype network except for haplotypes from Group V, which partitioned into two clusters. Although, no clear geographical pattern explains the separate clusters for Group V. Diversity measures were calculated for regional groups showed high haplotype diversity amongst all groups (Table 3).

### Differences in demographic histories

Estimates of Tajima's $D$ were significantly negative for all groups (mean Tajima's $D = -2.00$, $P = 0.01$), suggesting a selective sweep or demographic expansion for each group. Group V had the least negative value (Tajima's $D = -1.43$; $P = 0.04$), while Groups I (Tajima's $D = -2.23$; $P < 0.001$), III (Tajima's $D = -2.08$; $P = 0.001$), and IV (Tajima's $D = -2.26$; $P = 0.002$) had more negative values (Table S2). However, mismatch analyses of mitochondrial haplotypes under a model of spatial expansion showed different distributions for each group. Group I had a unimodal distribution, while Group III had a slightly bimodal distribution. Group IV had a sloped distribution while Group V had a pronounced bimodal distribution (Fig. 3).

### Patterns of isolation by distance

A Mantel test including all samples showed a weak (r = 0.047) but significant ($P = 0.002$) pattern of IBD when considering pairwise raw genetic distances and Haversine geographic distances. However, the greatest density of pairwise comparisons were centered at approximately 400 Km and 0.02 genetic distance (Fig. 4). Therefore, we performed independent Mantel tests for each identified group. Groups I (r = −0.026, $P = 0.894$) and IV (r = 0.026, $P = 0.321$) showed no significant pattern of isolation by distance, while Groups III (r = 0.137, $P = 0.001$) and V (r = 0.101, $P = 0.001$) showed weak but significant pairwise relationships between geographic and genetic distances. A moderate and

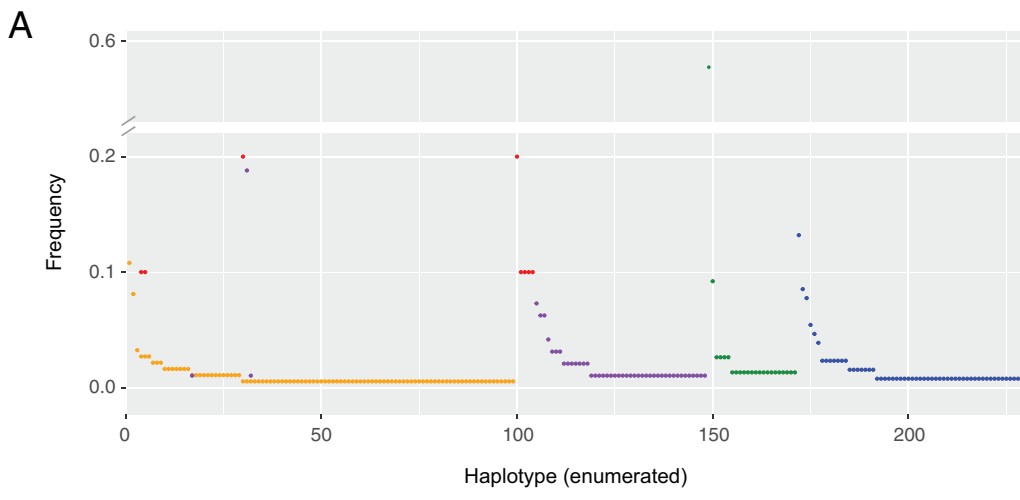

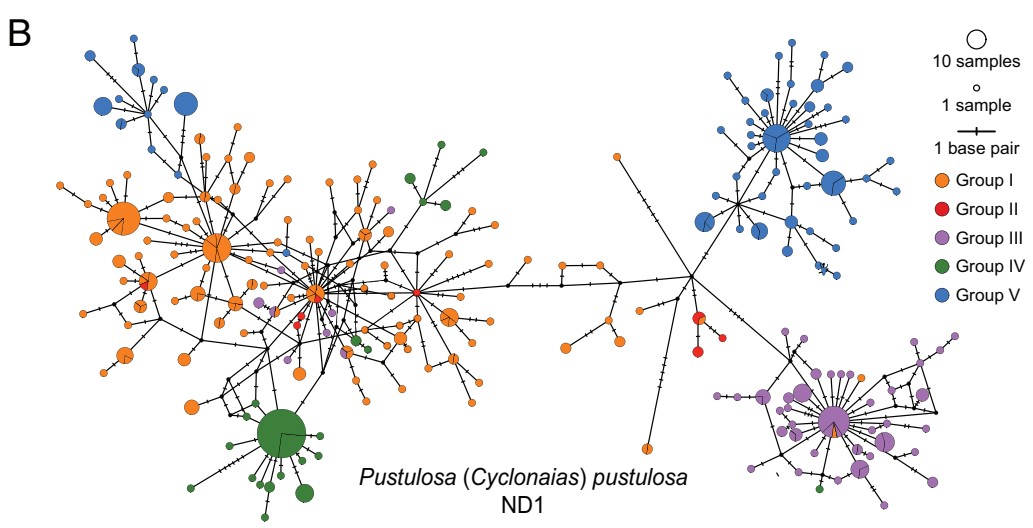

**Figure 2 Visualization of ND1 haplotype frequency and diversity in *Pustulosa* (*Cyclonaias*) *pustulosa*.** (A) Relative haplotype frequencies showing minimal haplotype sharing between groups. (B) Median-joining haplotype network where haplotypes are colored by their designated geographic group.

**Table 3 Measures of genetic diversity at the ND1 locus among designated populations of *Pustulosa* (*Cyclonaias*) *pustulosa*.**

| Group | Basins | $N$ | $H$ | $k$ (SD) | $\pi$ (SD) | SSF | GD (SD) |
|---|---|---|---|---|---|---|---|
| I | Mississippi basin and the Great Lakes | 185 | 99 | 4.86 (2.38) | 0.010 (0.005) | 0.028 | 0.977 (0.005) |
| II | Pascagoula/Pearl | 10 | 8 | 5.13 (2.71) | 0.010 (0.006) | 0.140 | 0.956 (0.059) |
| III | Sabine, Neches, Trinity, and San Jacinto | 96 | 47 | 4.15 (2.31) | 0.008 (0.004) | 0.060 | 0.950 (0.014) |
| IV | Brazos and Colorado | 76 | 23 | 2.51 (1.51) | 0.005 (0.003) | 0.349 | 0.649 (0.062) |
| V | Guadalupe, San Antonio, and Nueces | 129 | 58 | 6.63 (3.49) | 0.013 (0.007) | 0.045 | 0.963 (0.008) |

**Note:**
$N$, sample size; $H$, number of haplotypes; $k$, mean number of base pair differences; $\pi$, nucleotide diversity; SSF, sum of square frequencies; GD, gene diversity; and SD, standard deviations.

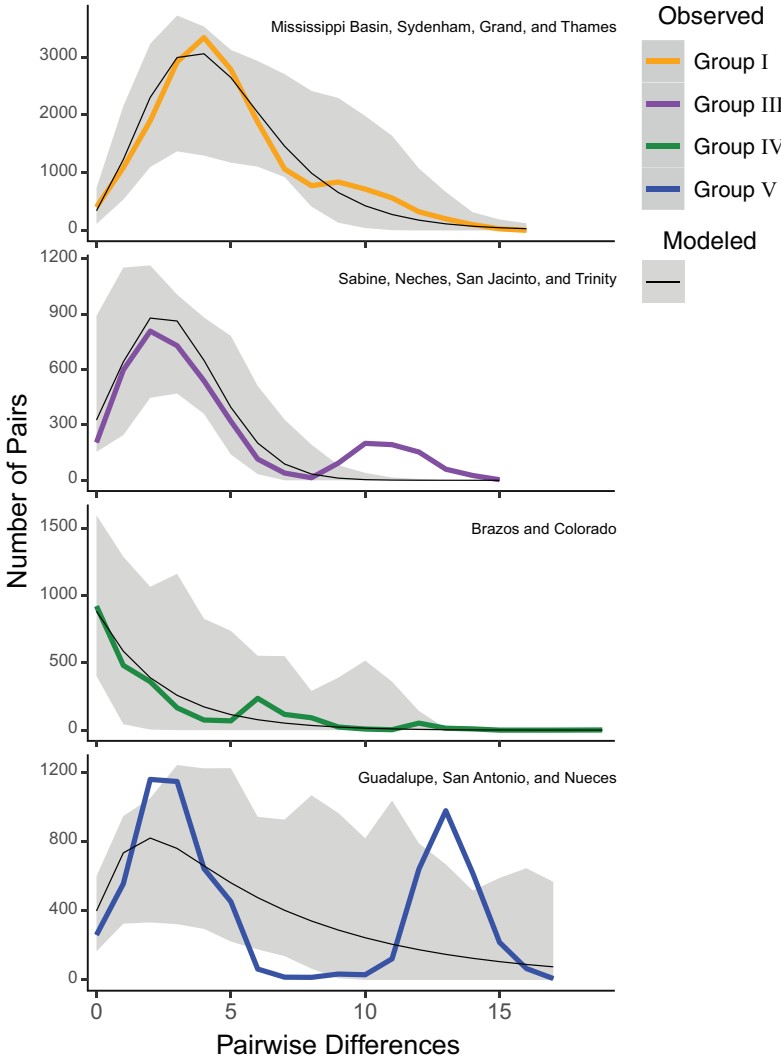

**Figure 3 ND1 mismatch distributions for *Pustulosa* (*Cyclonaias*) *pustulosa* regional groups.** Distributions show differing demographic histories among groups. Modeled frequencies (black line) and 95% confidence intervals (grey area) are also included.

significant IBD pattern was detected among Texas *P. pustulosa* populations that were included in Groups III, IV, and V (r = 0.44, *P* = 0.001) (Fig. 4).

## DISCUSSION

Species delimitation tests have shown that previously described species of *Cyclonaias* are synonyms of *C. pustulosa* (currently *P. pustulosa*) owing to the lack of monophyly and minimal genetic distances (*Johnson et al., 2018*; *Lopes-Lima et al., 2019*). However, different morphotypes are evident within the group and are associated with geography (*Lopes-Lima et al., 2019*), yet those differences might be owing to ecophenotypic plasticity or other factors not associated with genetic differentiation (*Inoue et al., 2013*). Therefore, we evaluated *P. pustulosa* sensu *lato* within a population genetics framework to inform future management efforts by potentially identifying RMUs based explicitly on population

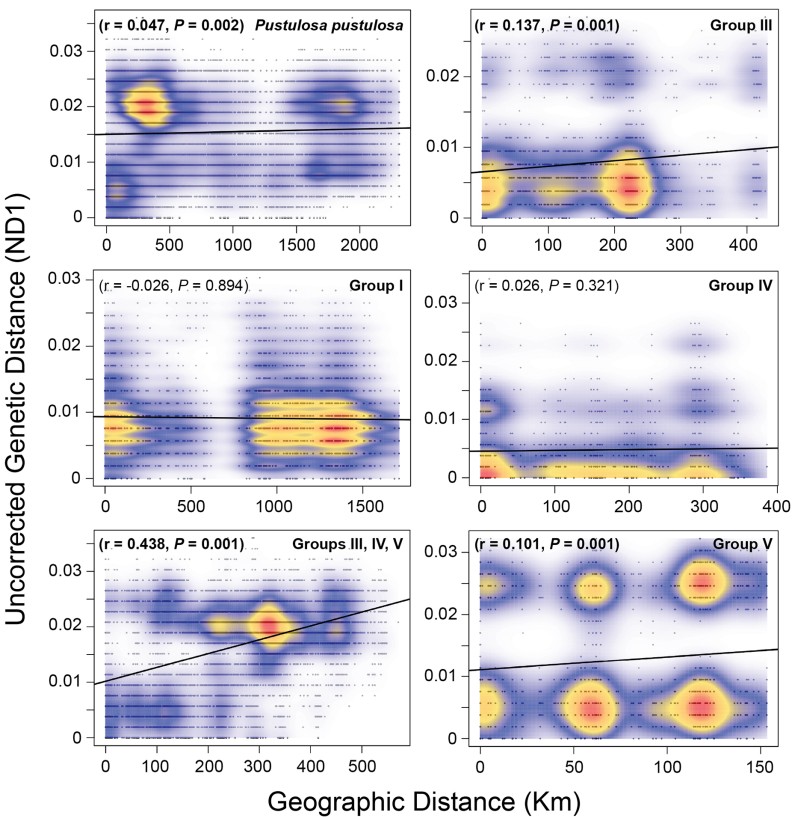

**Figure 4 Mantel tests for patterns of isolation by distance among *Pustulosa* (*Cyclonaias*) *pustulosa* individuals.** Plots showing the relationship between uncorrected pairwise genetic distances (ND1) and Haversine geographical distances (Km) for *P. pustulosa* populations and designated groups. High (red) to low (blue) correlation densities are overlaid on the pairwise comparisons along with the best-fit regression line (black).

structure. Mitochondrial DNA (ND1) showed that geographic groups of *P. pustulosa* sensu *lato* exhibit population differentiation (high pairwise $\Phi_{ST}$, Table S3) in addition to variable historical demographies illustrated by differing mismatch distributions, which has consequences for their conservation and management. Localities (*i.e.*, river drainages) genetically nested into regional groups, which were generally congruent to the distribution of previously described species within *P. pustulosa* sensu *lato* (Fig. 1); namely, *C. pustulosa* (Group I), *C. mortoni* (Group III), *C. houstonensis* (Group IV), and *C. aurea* (Group V) (*Lopes-Lima et al., 2019*).

The ND1 sequence data indicate *P. pustulosa* populations from rivers draining into the Mississippi and the Great Lakes (Lake Erie basins) experienced high gene flow historically, which is evidenced by low population differentiation among river drainages (Fig. 1) and a lack of a pattern of IBD (Fig. 4) despite the large distances between these populations. This is likely due to freshwater connectivity within the Mississippi basin and, we speculate, a post-glacial radiation into the Lake Erie basin. In contrast, other studies have found more population differentiation among freshwater mussels from the Mississippi basin and the Great Lakes; namely *Obovaria olivaria* (*Bucholz et al., 2022*), *Elliptio dilatata* and *Actinonaias ligamentina* (*Elderkin et al., 2008*), *Amblema plicata* (*Elderkin et al., 2007*),

and *Cumberlandia monodonta* (*Inoue et al., 2014*). However, some key differences in these species compared to *P. pustulosa* include smaller ranges or sampling sites that do not extend into Texas, and much lower haplotype diversity, which limit direct comparisons.

Populations of *P. pustulosa* from rivers within Texas draining into the Gulf of Mexico likely had lower connectivity historically as they show higher estimates of $\Phi_{ST}$ when compared to each other. Mantel tests of populations within groups showed no significant IBD for populations in Group IV (Brazos and Colorado), but Groups III and V showed weak but significant patterns of IBD. However, when analyzed together, populations within Groups III, IV, and III showed a moderate IBD pattern that demonstrates the contrasting population dynamics of *P. pustulosa* within Texas when compared to the rest of the range (Fig. 4). The rivers in Texas that were sampled during this study are terminally connected *via* saltwater, which acts as a dispersal barrier to freshwater mussels and likely limited gene flow between their populations despite the ability to interbreed, shared ancestry, relative geographic proximity, and high host vagility. Thus, lower connectivity between regions of Texas may explain the higher levels of differentiation found within Texas populations compared to lower differentiation between Mississippi/Great Lakes populations included in this study.

Among the groups that were delimited, estimates of Tajima's D and mismatch distribution analyses using ND1 data showed variation in population demography. Group V (Guadalupe–San Antonio) exhibited a more ragged distribution indicating stable historical populations, while Groups I (Mississippi–Great Lakes), III (Sabine, Neches, Trinity, and San Jacinto), and IV (Brazos–Colorado) showed primarily unimodal distributions suggesting historical bottlenecks and subsequent demographic expansions, likely at different times given the differences in mode (Fig. 3). These results imply that *P. pustulosa* in the Mississippi river basin may represent a recent colonization during the Holocene. Thus, shared haplotypes between Texas populations and Group I populations may result from shared ancestry. Alternatively, some haplotype sharing (Fig. 2A) may have resulted from misidentifications or recent translocations through anthropogenically-mediated migration, which has been documented in several other mussel species (*De Greef, Griffiths & Zeeman, 2013*; *Hewitt, Woolnough & Zanatta, 2019*; *Hoffman, Morris & Zanatta, 2018*; *Minchin, Maguire & Rosell, 2003*; *Zhulidov et al., 2005*). However, multi-locus nuclear markers and range-wide model-based analyses of population structure of *P. pustulosa* are still needed to disentangle the possibility of viable translocations between distant drainages.

*Pustulosa pustulosa* sensu *lato* may comprise a single species based on phylogenetic delimitation methods (*Johnson et al., 2018*; *Neemuchwala et al., 2023*), but within its extensive range distinct populations are associated with geography (*Lopes-Lima et al., 2019*). Data from our study suggest that each previously recognized nominal species within *P. pustulosa* in Texas represent separate populations, which is supported by some morphological differences and geographic separation (*Lopes-Lima et al., 2019*). Some individuals previously identified as *C. refulgens* clustered separately in the haplotype network, while others clustered within Group I haplotypes and likely represent misidentifications owing to potential phenotypic plasticity and their geographic proximity
(Fig. 2). However, only a few reference samples (*n* = 10) were found for populations from the Pascagoula/Pearl rivers and future investigations should increase population-level sampling in Louisiana and Mississippi and other undersampled areas (*i.e.*, central Mississippi basin). Increased sampling may reveal additional geographically partitioned groups. For example, haplotypes from Group V form two separate clusters in the haplotype network (Fig. 2B), yet they are not associated by river or locality, which also warrants further investigation to determine the ecological and evolutionary processes that might have led to this peculiar pattern.

### Conservation implications

The four major groups within *P. pustulosa* identified in this study may represent suitable RMUs for future conservation planning and actions, which is important given that different populations within its range are classified as Vulnerable (S3), Imperiled (S2), or Critically Imperiled (S1) (*Cyclonaias pustulosa*; NatureServe, 2022). Population genetics studies on other unionid mussel species have also suggested considering genetically distinct populations in management strategies (*Froufe et al., 2014*; *Geist et al., 2018*; *Skidmore et al., 2010*). These analyses suggest each of these major regional groups have unique, albeit recent (*i.e.*, after the Last Glacial Maximum, ~20,000 years ago), evolutionary trajectories and likely represent distinct ESUs. Therefore, assisted migration between regional groups should be avoided to prevent potentially disrupting adaptive gene complexes (*Allard, 1988*) or narrow larval mussel (glochidia)-host fish associations (*Modesto et al., 2018*; *Neemuchwala et al., 2023*). If assisted migration is warranted to augment overall population viability or mitigate local extirpations within RMUs, then management strategies should consider source populations that are within the RMU of the imperiled population(s). If this is not possible, then source populations should be selected from a RMU with the greatest genetic similarity to the sink populations.

### CONCLUSIONS

Overall, results from this study have shown that course-scale phylogenetic treatments of recently diverged, wide-ranging freshwater mussels may not reveal relevant fine-scale population structure, and that genetic data should also be analyzed within a population genetics framework. We sought to find similar haplotype frequency patterns in other species and organisms. However, in our literature search, we encountered a dearth of population genetic studies (*i.e.*, below the species level) encompassing a similar sampling regime for aquatic organisms that included rivers from Texas, the Mississippi, and Great Lakes basins in a single study. Most studies provide valuable phylogeographic treatments of related aquatic species or species delimitation within closely related groups (*e.g.*, Black Bass, see *Kim, Bauer & Near, 2022*). Yet, one recent study on Alligator gar (*Atractosteus spatula*), which included samples from Texas, the mid and lower Mississippi basin, and Florida, used both microsatellites and mitochondrial control region haplotypes to infer the population genetics of this species (*Bohn et al., 2023*). Based on control region haplotypes, they found higher differentiation among populations in Texas (average $\Phi_{ST}$ = 0.20, not

including the Red River) than among populations within the Mississippi basin (average $\Phi_{ST}$ = 0.17; including the Red River), and higher differentiation between Texas populations and those within the Mississippi basin (average $\Phi_{ST}$ = 0.28) (see *Bohn et al. (2023)*, Table S4). While consistent, the patterns of differentiation are not as prominent as in *P. pustulosa*. This could be attributed to the high salt tolerance exhibited by some Alligator gar, and thus, higher gene flow between river basins would be expected as salinity would not act as a strong barrier to gar as it would for freshwater mussels.

Texas Rivers were characterized by more population subdivision (higher estimates of pairwise $\Phi_{ST}$) among *P. pustulosa* populations when compared to rivers within the Mississippi drainage and rivers associated with the Great Lakes. Such high differentiation among river basins in Texas has been shown by previous ecological and genetic analyses of mussels and fish, which mussels need for large-scale dispersal during their parasitic larval stage on host fish. For example, Spotted Bass showed large genetic divergence between river basins (*Lutz-Carrillo et al., 2018*), and mussel and fish community composition were also found to differ significantly between river basins of Texas (*Dascher et al., 2018*). Yet, more studies of population subdivision among North American freshwater mussels and other aquatic species with distributions that span multiple basins that drain into the Gulf of Mexico (*e.g.*, *Bohn et al., 2023*) are still critically needed, which we begin to address in this study.

## ACKNOWLEDGEMENTS

We thank T. Marshall, G. Solis, M. Escandon, and C. Baca for assistance in the lab. We thank A. Tarter, B. MacVeigh, M. Sheldon, and M. Gibson for field assistance. We thank I. Porto-Hannes and U. Smart for comments on an earlier version of this manuscript.

### Funding

This work was supported by the Texas Department of Transportation's Research and Technology Implementation Division project agreement (No. 0-6882). There was no additional external funding received for this study. The funders had no role in study design, data collection and analysis, decision to publish, or preparation of the manuscript.

### Grant Disclosures

The following grant information was disclosed by the authors:
Texas Department of Transportation's Research and Technology Implementation Division: 0-6882.

### Competing Interests

The authors declare that they have no competing interests.

## Author Contributions

- David Rodriguez conceived and designed the experiments, analyzed the data, prepared figures and/or tables, authored or reviewed drafts of the article, and approved the final draft.
- Stephen F. Harding conceived and designed the experiments, performed the experiments, analyzed the data, prepared figures and/or tables, authored or reviewed drafts of the article, and approved the final draft.
- Shashwat Sirsi performed the experiments, analyzed the data, prepared figures and/or tables, and approved the final draft.
- Kelly McNichols-O'Rourke conceived and designed the experiments, performed the experiments, authored or reviewed drafts of the article, and approved the final draft.
- Todd Morris conceived and designed the experiments, performed the experiments, authored or reviewed drafts of the article, and approved the final draft.
- Michael R. J. Forstner conceived and designed the experiments, analyzed the data, prepared figures and/or tables, and approved the final draft.
- Astrid N. Schwalb conceived and designed the experiments, performed the experiments, authored or reviewed drafts of the article, and approved the final draft.

## Field Study Permissions

The following information was supplied relating to field study approvals (*i.e.*, approving body and any reference numbers):

Texas Parks and Wildlife Department and Canada Department of Fisheries and Oceans.

## DNA Deposition

The following information was supplied regarding the deposition of DNA sequences:

The ND1 sequences used in this study are available in the Supplemental File. The newly generated sequences are available at GenBank: OQ656460 to OQ656625.

## Data Availability

The aligned and trimmed DNA sequences are available in the Supplemental File.

## Supplemental Information

Supplemental information for this article can be found online at http://dx.doi.org/10.7717/peerj.15974#supplemental-information.

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
