# Peer review of "Mitochondrial sequence data reveal population structure within Pustulosa pustulosa"

_PeerJ, doi:10.7717/peerj.15974_

## Round 0.1 · original submission · Major Revisions

I concur with the comments of reviewer two and recommend the authors perform additional analyses and expand the discussion section by including possible explanations for the observed patterns of haplotypes that are inconsistent with geography. Also, the authors should respond to the comments from both reviewers.

Reviewer 1 ·

Basic reporting

Rodriquez et al. 2023 uses a population genetics framework to uncover RMU or potential management units for Cyclonaias species in Texas. While I think the study does a fine job with addressing population structure within and between major rivers/basins, it is wholly lacking any spatial examination of population structure. I feel that this paper would benefit greatly by using either Euclidean (geographic) or preferably true river distances (there are R packages for this, see riverdist, Tyers and Tyers 2017) to look for isolation-by-distance patterns within species, or even within the Texas drainages. I think this would round out the paper and help readers visualize the population structure in space, e.g. do populations follow strong patterns of IBD, and if they don’t, why might they not. Overall, I think the figures are crisp and clean to look at, and I take no issue with their analyses. I largely agree that freshwater mussels should, in general, be managed at the river or even population level, as they often have strong genetic strong between rivers/populations. Reading and incorporating some additional literature on other freshwater mussel population genetics may help the author flesh out some of these statements more.

Tyers, M., & Tyers, M. M. (2017). Package ‘riverdist’. Tech. rep., available at: https://cran.r-project.org/web/packages/riverdist/riverdist.pdf

Experimental design

I feel like this article could benefit from an examination of Isolation-by-Distance within this species.

Validity of the findings

no comment

Additional comments

L34: “Several aquatic species have wide distributional ranges in North America, yet few have been characterized within a population genetics framework.”

This statement is broad and untrue. Many aquatic species have been studied in a population genetic framework in North America, particularly fish and freshwater mussel species. See citations in comments below. Please remove this statement.

L64: “Previously detected genetic and geographic associations may represent undescribed intraspecific population structure within C. pustulosa sensu lato, which we consider to include the four previously recognized nominal species and C. pustulosa.”

What are these previously detected genetic and geographic associations? It would help the reader to expound here.

L211: “different morphotypes are associated with geography”

Species often have different morphotypes associated with geographic areas, however this does not mean those differences cannot be attributed to some phenotypic plasticity or epigenetic/environmental impacts. I would maybe expand here to say that morphological variation can be present within species with the absence of genetic dissimilarity, and there are a variety of reasons why this can occur (see Inoue et al. 2013).

Inoue, K., Hayes, D. M., Harris, J. L., & Christian, A. D. (2013). Phylogenetic and morphometric analyses reveal ecophenotypic plasticity in freshwater mussels O bovaria jacksoniana and V illosa arkansasensis (B ivalvia: U nionidae). Ecology and Evolution, 3(8), 2670-2683.

L222: “ND1 sequence data indicate C. pustulosa populations from rivers draining into the Mississippi and the Great Lakes (Lake Erie basins) are functionally connected”

I would remove this statement. ND1 is mtDNA locus, and mussels are long-lived with long and overlapping generations. This means that it can take several generations for any barriers to gene flow to be detected in real world populations. We can’t test if rivers are “functionally connected” for these species without understanding true migration.

Hoffman, J. R., Willoughby, J. R., Swanson, B. J., Pangle, K. L., & Zanatta, D. T. (2017). Detection of barriers to dispersal is masked by long lifespans and large population sizes. Ecology and Evolution, 7(22), 9613-9623.

L266: “Population genetics studies on other unionid mussel species have also suggested considering genetically distinct populations in management strategies”

This may be an excellent place for the author to try to weave in some more ideas from existing population genetic literature on freshwater mussel species and speak to the fact that it is very common for mussels to have high Fst between rivers/regions and how this relates to management implications (e.g. try to pick from nearby populations whenever possible).

L 283: “dearth of population genetic literature on aquatic species (i.e. below the species level)”

I am somewhat at a loss the author means here, as this statement is quite frankly untrue. I have listed numerous papers below which are just a small fraction of the current literature. Please remove this statement. There is also a general lack of citation for existing population genetic literature on freshwater mussel species throughout this article, that being said it is not necessary to cite an abundance of literature.

See:

Elderkin, Curt L., et al. "Population genetics and phylogeography of freshwater mussels in North America, Elliptio dilatata and Actinonaias ligamentina (Bivalvia: Unionidae)." Molecular Ecology 17.9 (2008): 2149-2163.

Elderkin, C. L., Christian, A. D., Vaughn, C. C., Metcalfe-Smith, J. L., & Berg, D. J. (2007). Population genetics of the freshwater mussel, Amblema plicata (Say 1817)(Bivalvia: Unionidae): evidence of high dispersal and post-glacial colonization. Conservation Genetics, 8, 355-372.

Bucholz, J. R., Sard, N. M., VanTassel, N. M., Lozier, J. D., Morris, T. J., Paquet, A., & Zanatta, D. T. (2022). RAD‐tag and mitochondrial DNA sequencing reveal the genetic structure of a widespread and regionally imperiled freshwater mussel, Obovaria olivaria (Bivalvia: Unionidae). Ecology and Evolution, 12(1), e8560.

Whelan, N. V., Galaska, M. P., Sipley, B. N., Weber, J. M., Johnson, P. D., Halanych, K. M., & Helms, B. S. (2019). Riverscape genetic variation, migration patterns, and morphological variation of the threatened Round Rocksnail, Leptoxis ampla. Molecular ecology, 28(7), 1593-1610.

Abernethy, E., McCombs, E., Siefferman, L., & Gangloff, M. (2013). Effect of small dams on freshwater mussel population genetics in two southeastern USA streams. Freshwater Mollusk Biology and Conservation, 16(1), 21-28.

Inoue, K., Lang, B. K., & Berg, D. J. (2015). Past climate change drives current genetic structure of an endangered freshwater mussel species. Molecular Ecology, 24(8), 1910-1926.

Inoue, K., E. M. Monroe, C. L. Elderkin, and D. J. Berg. "Phylogeographic and population genetic analyses reveal Pleistocene isolation followed by high gene flow in a wide ranging, but endangered, freshwater mussel." Heredity 112, no. 3 (2014): 282-290.

Zieritz, A., Hoffman, J.I., Amos, W. and Aldridge, D.C., 2010. Phenotypic plasticity and genetic isolation-by-distance in the freshwater mussel Unio pictorum (Mollusca: Unionoida). Evolutionary Ecology, 24, pp.923-938.

Shelley, J. J., Holland, O. J., Swearer, S. E., Dempster, T., Le Feuvre, M. C., Sherman, C. D., & Miller, A. D. (2022). Landscape context and dispersal ability as determinants of population genetic structure in freshwater fishes. Freshwater Biology, 67(2), 338-352.
L295: This would be a good place to cite some existing studies on freshwater mussel/fish population genetics.

Table 3. It would really help to have the name of each group on this table, as it’s not entirely intutitive which groupings the author is using here. I assume it’s the biogeographical provinces? But simply listing them would make this more clear.

Reviewer 2 ·

Basic reporting

The goal of the research was to investigate the genetic variation and demographic history with the freshwater mussel Cyclonaias pustulosa sensu lato. The motivation for this study is to further inform the management of freshwater mussels in through the delimitation of genetic/geographic units. To accomplish this the authors assemble a data set of mitochondrial DNA sequences of the first subunit of NADH dehydrogenase gene and examine the relationships and distribution of haplotypes as well as investigate the signature of demographic history (e.g. population expansion) in the data.

The authors added a substantial number of samples and sequences to existing DNA sequences in GenBank.

The main results of this research are not substantially different from and earlier paper by Johnson et al. 2018, which included a nuclear genetic marker in addition to a large mitochondrial data set that is similar in geographic scope to the one presented here.
The authors intimate that their approach includes more of a population genetic framework, but the difference between the present study and Johnson et al 2018 are not substantial in my opinion.

I do not think the increased sample sizes and sample locations in this study result in a substantial enough increase in knowledge over that study to warrant publication.

If the authors could add additional analyses that take advantage of the added sample to further test explicit demographic hypotheses and further develop the management implications based on these.

The results reveal patterns in some haplotypes that are not consistent with geography – which is atypical of most haplotypes in the data set – but the authors do not explore these results further or advance any possible explanations. I think addressing the possible causes of this pattern would enhance the paper greatly.

Experimental design

No Comment

Validity of the findings

No Comment

Additional comments

No Comment

---

## Round 0.2 · Minor Revisions

I think the authors have addressed the comments on the previous version, and the manuscript is much improved.

Reviewer 1 ·

Basic reporting

I think the authors did a good job addressing reviewer comments. I take no issue with the manuscript as it stands now. My only final remark is that now Cyclonaias pustulosa falls under Pustulosa (Neemuchwala et al. 2023). Authors should use the updated taxonomy.

Experimental design

The additional IBD analyses address my prior comments.

Validity of the findings

While I do not feel that this article is substantially different from prior work in Cyclonaias, I do think the additional data provided are a positive contribution to existing data for this species.

---

## Round 0.3 · accepted · Accept

I recommend the manuscript be accepted for publication.